# Evolutionary game study on multi-agent value co-creation of service-oriented digital transformation in the construction industry

**Shiming Wang[1], Hui Su[1]\*, Qiang Hou[2]**

**1** School of Business Administration, Liaoning Technical University, Huludao, China, **2** School of Management, Shenyang University of Technology, Shenyang, China

\* 1984723125@qq.com

**Data Availability Statement:** All relevant data are within the paper and its Supporting Information files.

**Funding:** This work was supported in part by the Basic Scientific Research Project of Education Department of Liaoning Province under grant

## Abstract

The service-oriented digital transformation of the construction industry is a development trend of cross-border industrial integration and transformation and upgrading in the digital economy environment, and collaborative value creation among stakeholders is seen as a strategic imperative to promote this process. This study aims to achieve efficient collaborative value co-creation and accelerate the digital transformation process of the construction industry by exploring the collaborative strategies and evolution laws of value co-creators in the digital service ecosystem of the construction industry. Based on evolutionary game theory and methods, this paper analyzes the evolutionary stability strategies and conditions of each participant in the service-oriented value chain at different stages of the digital transformation of the construction industry. It is found that with the improvement of the level of digitalization, the degree of cooperation among game players continues to increase until a stable state of full cooperation is achieved. The initial willingness of the game players to cooperate accelerates the speed of the system's evolution to the stable state of full cooperation in the middle stage of digital transformation. Additionally, the improvement of the construction process digitalization level can subvert the evolution result of full non-coordination caused by a low initial willingness to cooperate. The research conclusions and corresponding countermeasures and suggestions can provide a strategic reference for the service-oriented digital transformation of the construction industry.

## 1. Introduction

As the pillar industry of the national economy, the construction industry continues to face the leading problem of "being large but not strong". Under the traditional project construction mode, the coordination among owners, design units, and construction units has been recognized as a hindrance to the development of the construction industry. With the arrival of Construction 4.0, emerging digital technologies are inevitable in the development of the construction industry to achieve the comprehensive coordination of approval and decision-making, planning and design, construction operation and maintenance. Digital technology

LJKMR20220709 and in part by the Social Science Fund of Liaoning Province under grant L21BGL027. The funders had no role in study design, data collection and analysis, decision to publish, or preparation of the manuscript.

**Competing interests:** The authors have declared that no competing interests exist.

can integrate customer value into the process of enterprise product and value creation, which is a prominent feature of digital transformation and an important breakthrough for construction enterprises in achieving the integration of digitalization and service. The 14th Five-Year Plan for the Development of the Digital Economy clearly indicated that digital services are a new trend in the development of China's digital economy and established the accelerated integration of industrial digital transformation and productive services as the goal during the 14th Five-Year Plan period. Driven by both digital technology and policy, the service-oriented digital transformation of the construction industry has not only injected new vitality into the traditional construction industry but also provided new opportunities to increase the value of various stakeholders by transforming from product construction to service construction with the help of digital technology. This has become the key to the high-quality development of the construction industry. Specifically, in the construction industry, all participants take the needs of owners (customers) as the starting point in the construction process and cultivate a new ecology of the construction industry through the integration of "the construction gene" and "digital gene". Construction enterprises are the core subject of industrial operations, and digital solution suppliers are the digital enablers. The two subjects work together to promote the full participation of customers by adding digital derivative services in the construction process and integrating decentralized heterogeneous resources to achieve the optimization of the entire production process. However, it is difficult for the independent digital investment of each subject to give play to the advantages of digital technology, and collaboration and communication among stakeholders are particularly important in resource integration, operation process adjustment, and other aspects [1, 2]. Xie *et al.* [3] pointed out that the social facts hidden behind digital transformation and the essence of digital resources are cooperation and sharing, and collaboration is crucial for enterprises in the transformation to stand and develop in the turbulent market environment. In addition, after the impact of COVID-19, the traditional production mode and development path of construction enterprises have been severely damaged. It is urgent to cooperate to obtain and integrate more external resources and revitalize the industry market through the new business model. Obviously, exploring the evolutionary laws of collaborative strategies of value co-creators in the service-oriented digital transformation of the construction industry has become a necessary entry point for promoting the efficient transformation of the construction industry and restoring market vitality.

Different from previous studies, this article takes the digital service ecosystem of the construction industry as the research object, focusing on the collaborative behavior among value co-creators in the value chain, and incorporating digital level factors that have not been addressed in existing research into the research scope, to preliminary explore the integration of digital and service-oriented in the construction industry. Considering that there are multiple strategic combinations of value co-creation behaviors among various entities in realistic situations, it may not be a coordinated state of reaching consensus and cooperation. From the perspective of value co-creation, this study constructs an evolutionary game model for three parties, namely, construction enterprises, digital solution suppliers, and customers. It analyzes the dynamic evolution process of strategy selection of each participant in the service-oriented value supply chain and simulates the evolution results of value co-creation collaborative strategy selection under the change of factors. It has certain practical guiding significance for grasping the essential characteristics and operation mechanism of value co-creation in the digital transformation process of the construction industry, and scientifically and accurately implementing countermeasures based on it to promote the high-quality development of the construction industry.

Section 2 reviews the previous studies on service-dominant logic and digital transformation respectively, and summarizes the research gaps. Section 3 analyzes the structure of the digital

service ecosystem in the construction industry and defines the collaborative mechanism and role orientation of value co-creation of the multi-agent in the service-oriented digital transformation value chain, and constructs an evolutionary game model of value co-creation in the digital service ecosystem of the construction industry. Section 4 presents the stability analysis results of the evolutionary game model. Section 5 through digital simulation, reveals the dynamic evolutionary law of the game among value co-creators in the digital service ecosystem of the construction industry and discusses the role mechanism of the digital level in the evolutionary stability strategy. Section 6 concludes with a summary of the main results.

## 2. Literature review

In the relevant research on value co-creation, a service-oriented logic occupies a dominant position. Vargo and Lusch [4] understood value co-creation from the perspective of a service-oriented logic and believed that value is defined by consumers and co-created by the joint production process of enterprises and consumers. This concept shows that the interaction between enterprises and consumers is not only a simple business transaction but also a form of process-oriented cooperation. The binding relationship between enterprises and consumers in the early, middle, and late stages of production is not only an effective means for enterprises to capture competitive advantages but also improves consumers' satisfaction with service quality [5, 6]. Wang *et al.* [7] analyzed the problems in the Engineering-Procurement-Construction (EPC) project management of the construction industry from the perspective of a service-oriented logic and proposed a specific path to overcome the dilemma of low levels of trust between the contracting parties through value co-creation. With the deepening of the development of a service-oriented logic to service ecosystem theory, the focus of value co-creation has gradually expanded from the early binary interaction between enterprises and consumers to the diversified interaction within the ecological network system [8]. The service ecosystem combines the research perspective of the ecosystem and takes the service-oriented logic theory as the core. It is a dynamic system with multi-agent service exchange, value co-creation, and self-regulation functions [9]. The collaborative strategy of participants is the key to maintaining the stability and sustainable development of the service ecosystem [10]. Pacheco *et al.* [11] pointed out that the effective tools for studying cooperative behavior are evolutionary game method and replication dynamics mechanism. Throughout the literature on the game relationship between participants in the service ecosystem, most studies focus on the multilevel interaction of participants. The service ecosystem carries out resource integration and service exchange at the micro, meso, and macro levels. At the micro level, the binary interaction between enterprises and consumers has been a focus. For example, Sun *et al.* [12] built a symbiotic evolutionary model of three value co-creation units, namely, third-party service platforms, service providers, and service consumers, and analyzed their symbiotic conditions to maintain stability in the service ecosystem. At the meso level, the interaction among enterprises has been emphasized. For example, Zhang *et al.* [13] further considered the value chain system composed of manufacturers, suppliers, logistics providers, and third-party manufacturing platforms and analyzed its trend toward collaborative evolution based on game theory. The macro level focuses on the introduction of social participants, including the state, culture, market, etc., and covers the micro and meso levels. For example, Li and Li [14] brought the government, enterprises, and consumers into the same system to explore the impact of government subsidy policies and consumer purchase behavior on the intelligent transformation decision-making of manufacturing enterprises. These studies further illustrate that game theory provides ideas for analyzing the logic behind the collaborative behavior of value co-creators. Specifically, game theory can guide the decision-making process of participants, while

replication dynamics can reveal the evolution of collaborative strategies in the process of digital transformation.

It has been noted that the digital transformation strategy should not only consider the principle and value of the coordination between society and the economy but also pay attention to the changes in the creation mechanism of products and services [15]. The development and upgrading of digital technology have reshaped the interaction boundary among the subjects of the service ecosystem and has brought more stakeholders into the process of value creation with low-cost communication and connections around customer needs, promoting the collaborative sharing of heterogeneous resources among multiple subjects [16–18]. It shows that the rapid development of digital technologies such as big data and the Internet of Things(IoT) has not only improved the complexity of the value co-creation network but also reshaped the value co-creation mode of the subjects of the service ecosystem, leading the service ecosystem to address the challenge of transforming to the digital service ecosystem. Kolagar *et al.* [19] systematically reviewed the trigger factors, driving factors, transformation stages, and the activities at each stage of the transformation to the digital service ecosystem and provided a specific research agenda for the different stages. At the same time, how to maintain the sustainable and stable development of the digital service ecosystem has become the focus of academic attention. Scholars generally believe that participants can promote a series of value co-creation activities to achieve the feasibility and stability of the system by integrating collaborative resources based on a consistent structure and seeking cooperation based on win-win thinking [20]. Goudarzi *et al.* [21] studied the behavior strategies of participants in the service ecosystem in the digital context, constructed cooperative and noncooperative game models between service providers and consumers in the cloud manufacturing service ecosystem, and found that the cooperative game is a win-win for both parties by comparing the values of service quality and the profit values of participants.

The above studies are all focused on the manufacturing industry, while in contrast to the construction industry, digital technologies represented by BIM, the Internet of Things, and cloud computing are changing the construction process and business model of the traditional construction industry, driven by the dual drive of Industry 4.0 and industry digitization. For example, cloud computing enables architectural designers and workers to work and operate remotely, quickly optimizing and restructuring the construction supply chain [22]; In order to further strengthen information models such as BIM, CPS (Cyber Physical System) has been proposed and widely used for the installation control of prefabricated formwork, risk control of blind plate cranes, and optimization of safety management during the construction process [23]. In addition, the service-oriented innovative economy is becoming a new trend in the construction industry. In the service ecosystem, the service-oriented construction industry realizes value co-creation through the reconstruction of the value chain among heterogeneous entities, and the cooperation between the construction enterprises and the upstream and downstream enterprises of the value chain can optimize the efficiency of resource allocation [24]. Therefore, the collaborative behavior of the various entities in the value chain is very important for the value co-creation of the service ecosystem in the construction industry. The emergence of digital technology provides a convenient and effective new means for the cooperation between various subjects. Prebanić and Vukomanović [25] have found that digital technologies such as BIM and virtual reality can effectively attract owners to participate in the design and development process and improve the communication efficiency and collaboration level between owners and construction stakeholders by systematic review of stakeholder management practices in the construction industry. At the same time, several recent studies also focus on the interaction between digitalization and service in the construction industry and propose that digital technology can promote construction enterprises to provide services for

end customers, such as providing customers with preventive services and more value through data collection. In contrast, the creativity and inspiration of digital technology innovation can also be obtained from market solutions by introducing service-oriented models [26, 27].

In summary, the value co-creation of the service ecosystem in the digital context has attracted extensive attention from the academic community. However, most studies focus on the manufacturing industry. Research on the integration of digitalization and services in the construction industry mainly focuses on promoting the service efficiency of construction enterprises and the entire service ecosystem through digitalization. Few studies focus on the collaboration of stakeholders in the value chain under the digital context. There is still room for breakthroughs in the integration research of digitalization and service in the construction industry. Therefore, in order to make up for the shortcomings of existing research, this paper proposes an evolutionary game model for value co-creation of the digital service ecosystem in the construction industry, exploring the evolutionary process and dynamic development trend of collaborative behavior among various participants in the process of digital service in the construction industry.

## 3. Multi-agent value co-creation evolutionary game model

### 3.1 Role recognition

With reference to the interactive model of participants in the digital transformation of the construction industry by Chen *et al.* [28], this paper proposes that the digital service ecosystem of the construction industry is a complex network system consisting of a service-oriented value chain at the core of the system and external auxiliary institutions (including universities and scientific research institutions, governments, financial institutions, and industry network platforms). The internal value chain, supported by external auxiliary institutions, meets the needs of customers through a clear division of labor and cooperation, integration, and sharing of data resources and aims to achieve value co-creation through systematic collaborative innovation [29]. The digital service ecosystem of the construction industry is shown in Fig 1.

Fig 1 shows that the main body of value co-creation in the service-oriented digital transformation of the construction industry consists of construction enterprises, digital solution suppliers, and customers. Construction enterprises are the key node and leading force of the whole value co-creation system. They use digital technology, digital products, and digital management in the whole life cycle of project construction to transform customers' needs into entities. Enterprises are not only the demanders of digital transformation but also the implementation terminals, driving and leading the digital transformation and coordinated development of the whole system. Overall, the construction industry is relatively decentralized, which also leads to the fragmentation of data storage in all stages of construction. The intelligent construction collaborative management platform based on project construction can effectively break the status of isolated data islands. As the developers of digital technology integration, digital software, and hardware products and services, digital solution suppliers are the key node in the value chain. Customers are the source of value creation for construction enterprises and digital solution suppliers in the service-oriented value chain, and their personalized needs and use of information feedback are the innovative force of the whole value co-creation system.

### 3.2 Tripartite game analysis and research hypothesis

1. Collaboration between construction enterprises and customers

In the context of digital transformation, customers are not the end of transactions but the nodes that create value for enterprises. With the help of digital technology, construction

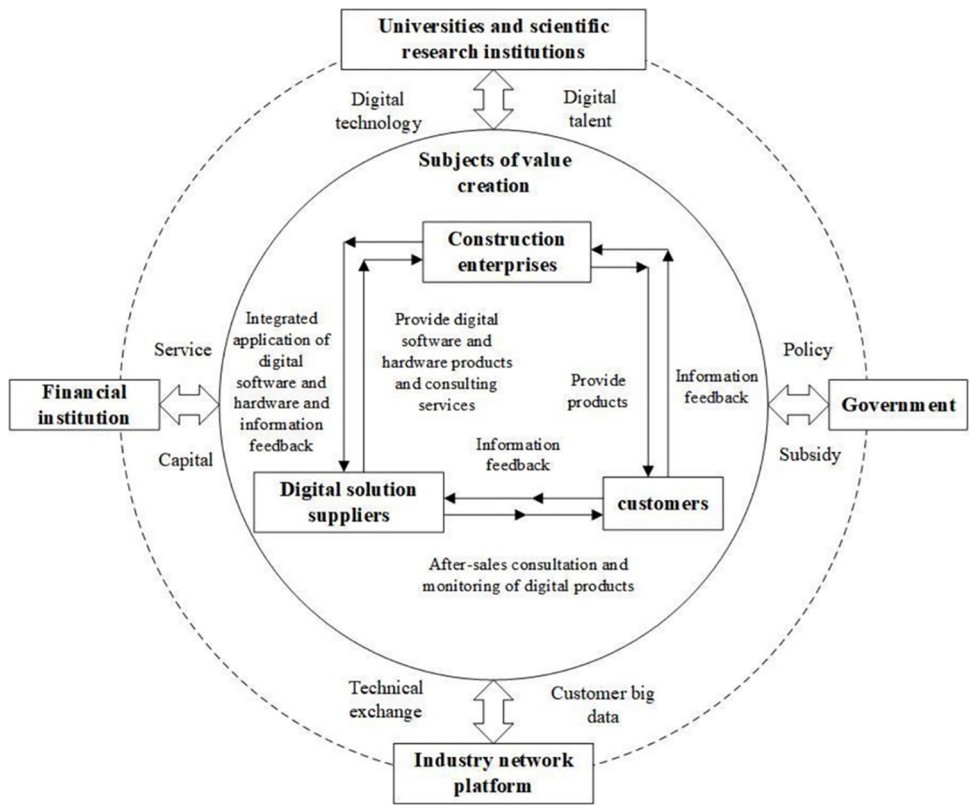

**Fig 1. Digital service ecosystem in the construction industry.**

enterprises enable customers to participate in all aspects of the whole life cycle of construction projects from planning, design, and construction to operation and maintenance and create platforms and opportunities for communication and feedback with customers. Through the feedback data provided by customers, construction enterprises can more clearly and directly guide how to meet their needs, thereby enhancing customer perceived value, achieving the goal of maximizing corporate profits, and expanding the profit space and scope. In addition, customers can combine the products or services provided by construction enterprises with their resources to stimulate their creative needs and create value through interaction with construction enterprises. However, customers participating in the collaborative process must bear a certain amount of energy and opportunity costs. If customers do not participate in collaboration, their own perceived value will be reduced and collaborative benefits will be lost. Meanwhile, construction enterprises will suffer information losses.

Based on the above analysis, the following theoretical hypotheses are proposed:

H1: Under the supervision of construction enterprises, customers will ultimately choose to participate in the collaboration.

2. Collaboration between construction enterprises and digital solution suppliers

The collaboration between construction enterprises and digital solution suppliers can improve the ability to collaborate and integrate resources between value co-creators to achieve a rapid response to changes, an accurate connection between supply and demand, a flexible allocation of resources, and more innovation possibilities. The whole value chain uses digital technology to realize efficient connections and optimize research and development design,

construction, operation management, marketing, and after-sales service through a data-oriented service model. At the same time, it can also deeply endow the wisdom and value of each element in the operation process of construction enterprises and promote the optimization of the whole service-oriented operation process and operational efficiency. In addition, according to resource-based theory, the integration and full utilization of resources is the main source for enterprises to maintain competitive advantages [30]. In the process of service-oriented digital transformation, service quality is an important factor affecting the core competitiveness of construction enterprises. Therefore, construction enterprises will also outsource their non-core professional digital after-sales service business to digital solution suppliers to provide more professional services. However, after signing the service outsourcing agreement, digital solution suppliers are likely to damage the interests of construction enterprises and customers by reducing the quality of digital software and hardware facilities, leaking information, and lowering the service level to maximize their own interests. To prevent these risks, construction enterprises need to impose certain regulatory measures on digital solution suppliers to improve their service quality. In this process, construction enterprises have to bear the cost of collaborative supervision, and digital solution suppliers have to bear the cost of collaborative innovation. If construction enterprises do not implement collaborative supervision strategies, due to the lack of coordination and communication with digital solution suppliers and customers, as well as the reduction of service quality of digital solution suppliers, construction enterprises and digital solution providers will lose some customers. If digital solution providers do not participate in collaborative innovation, construction enterprises may incur losses due to the lack of digital services support.

Based on the above analysis, the following theoretical hypotheses are proposed:

H2: Whether digital solution suppliers participate in collaborative innovation depends on the strategic choices of construction enterprises.

3. Collaboration between digital solution suppliers and customers

The collaboration between digital solution suppliers and customers is mainly achieved through feedback data provided by construction enterprises or direct feedback data from customers. At this point, the digital platform provides a bridge for digital solution suppliers to interact with customers, which helps digital solution suppliers quickly capture current market needs and personalized customer needs and improve their own and even the entire value chain's service innovation capabilities. In addition, digital solution suppliers can realize data access in the product after-sales service link with the help of digital platforms. Customers continuously provide feedback data to digital solution suppliers or construction enterprises in direct and indirect ways during the full cycle of services provided by digital solution suppliers, enhancing their ability to create value. If digital solution suppliers do not participate in collaborative innovation, customers will suffer certain losses due to the lack of support from digital services and the decrease in service quality. Similarly, if customers choose not to participate in collaborative innovation, digital solution providers will also lose the motivation for innovation due to a lack of information feedback.

Based on the above analysis, the following theoretical hypotheses are proposed:

H3: The customers' strategic choice is synchronized with the digital solution suppliers.

In addition, the level of digitalization affects the benefits and costs of construction enterprises and digital solution suppliers in the game process. The level of digitalization is reflected by the level of enterprise data collection and analysis, the level of the platform, and the digitalization level of the construction process [31]. First, data are an important information asset for enterprises to understand and develop demand markets and an important medium for

enterprises to communicate with customers and enter the ecosystem. Second, in the platform stage, with the help of the data sharing and connection of the construction end, the sales end, and the service end, the digital platform has realized the close connection among the value creation entities, greatly reducing the cost of cooperation and transactions among the entities. Finally, using digital technology to improve the efficiency of operations and construction can provide a new driving force for the digital transformation of construction enterprises and the efficient development of the digital service ecosystem of the construction industry.

Based on the above analysis, the following theoretical hypotheses are proposed:

H4: The digital level can promote the collaborative value co-creation of game players.

### 3.3 Basic assumptions of the model

Based on the above analysis of the synergy mechanism among value co-creators in the service-oriented digital transformation of the construction industry, construction enterprises, digital solution suppliers, and customers, as the main players of value co-creation, are the players in the game. They are all limited rational individuals with the ability to learn and adapt to dynamic environmental changes and adjust and optimize their strategies from their own interests. The strategy set of construction enterprises is {collaborative supervision, no collaborative supervision}, and the probability that construction enterprises choose the strategy of collaborative supervision at time $t$ is $x$; then, the probability that they choose the strategy of no collaborative supervision is $1-x$. The strategy set of digital solution suppliers and customers is {collaborate, not collaborate}. Where the probability of digital solution suppliers and customers choosing the "participate in collaboration" strategy at time $t$ is $y$ and $z$, respectively, then the probability of choosing the "not participate in collaboration" strategy is $1-y$ and $1-z$, respectively, $x,y,z \in [0,1]$, and both are functions of time $t$.

The initial revenues of construction enterprises, digital solution suppliers, and customers are $I_1$, $I_2$ and $I_3$, respectively. When all three parties participate in value co-creation, additional synergy benefits will be created, and the synergy benefits will be distributed proportionally among the participants [30]. The distribution coefficient of the collaborative income of construction enterprises is $\alpha$, that of digital solution suppliers is $\beta$, that of customers is $\gamma = 1-\alpha-\beta$, and $\alpha,\beta,\gamma \in [0,1]$. When construction enterprises, digital solution providers, and customers choose to participate in value co-creation, it is assumed that the collaboration cost that the three parties need to bear is $C$, and the collaboration cost will be distributed proportionally among the participants [32]. The collaborative cost-sharing coefficient of the construction enterprises is $\lambda$, the collaborative cost-sharing coefficient of the digital solution suppliers is $\eta$, the collaborative cost-sharing coefficient of the customers is $\sigma = 1-\lambda-\eta$, and $\lambda,\eta,\sigma \in [0,1]$. The losses caused to construction enterprises and digital solution suppliers due to their failure to choose collaborative supervision are $L_1$. The losses caused to construction enterprises and customers by digital solution suppliers choosing not to participate in collaborative innovation are $L_2$. The losses caused to construction enterprises and digital solution suppliers by customers choosing not to participate in collaborative innovation are $L_3$. The coefficients of the three factors reflecting the level of digitalization affect the benefits and costs of participants. The data collection and analysis coefficient is expressed as $\mu$, the platform coefficient is expressed as $v$, and the digitization of the construction process coefficient is expressed as $\omega$, and $\mu,v,\omega \in [0,1]$.

### 3.4 Construction of the evolutionary model

Based on the above assumptions, the evolutionary game income matrix of construction enterprises, digital solution suppliers, and customers can be constructed as shown in Table 1.

**Table 1. The payoff matrix of the tripartite game.**

| The strategies of subjects | | Customers participate in collaboration ($z$) | |
|---|---|---|---|
| | | Digital solution suppliers participate in collaboration ($y$) | Digital solution suppliers don't participate in collaboration ($1-y$) |
| Construction enterprises | Collaborative supervision ($x$) | $A_1 = (1+\omega)(I_1 + \alpha\Delta I) - (1-\mu-v)\lambda C$ | $A_2 = (1+\omega)I_1 - (1-\mu-v)\lambda C - L_2$ |
| | | $B_1 = (1+v+\omega)(I_2 + \beta\Delta I) - \eta C$ | $B_5 = I_2$ |
| | | $C_1 = [I_3 + (1-\alpha-\beta)\Delta I] - (1-\mu-v)(1-\lambda-\eta)C$ | $C_3 = I_3 - (1-\mu-v)(1-\lambda-\eta)C - L_2$ |
| | Not collaborative supervision ($1-x$) | $A_5 = (1+\omega)I_1 - L_1$ | $A_6 = (1+\omega)I_1 - L_1 - L_2$ |
| | | $B_2 = (1+\omega)I_2 - \eta C - L_1$ | $B_6 = I_2 - L_1$ |
| | | $C_2 = I_3 - (1-\lambda-\eta)C$ | $C_4 = I_3 - (1-\lambda-\eta)C - L_2$ |
| The strategies of subjects | | Customers don't participate in collaboration ($1-z$) | |
| | | Digital solution suppliers participate in collaboration ($y$) | Digital solution suppliers don't participate in collaboration ($1-y$) |
| Construction enterprises | Collaborative supervision ($x$) | $A_3 = (1+\omega)I_1 - \lambda C - L_3$ | $A_4 = (1+\omega)I_1 - \lambda C - L_2 - L_3$ |
| | | $B_3 = (1+v+\omega)I_2 - \eta C - L_3$ | $B_7 = I_2 - L_3$ |
| | | $C_5 = 0$ | $C_7 = 0$ |
| | Not collaborative supervision ($1-x$) | $A_7 = (1+\omega)I_1 - L_1 - L_3$ | $A_8 = (1+\omega)I_1 - L_1 - L_2 - L_3$ |
| | | $B_4 = (1+\omega)I_2 - \eta C - L_1 - L_3$ | $B_8 = I_2 - L_1 - L_3$ |
| | | $C_6 = 0$ | $C_8 = 0$ |

## 4. Stability analysis of the evolutionary game model

### 4.1 Model equilibrium analysis

According to the income matrix of the evolutionary game, the expected income $E_1(x)$ or $E_1(1-x)$ of construction enterprises when they choose the "cooperative supervision" or "no cooperative supervision" strategy are, respectively:

$$
\begin{aligned}
E_1(x) &= yzA_1 + z(1-y)A_2 + y(1-z)A_3 + (1-y)(1-z)A_4 \\
&= yz(1+\omega)\alpha\Delta I + yL_2 + z[(\mu+v)\lambda C + L_3] + [(1+\omega)I_1 \\
&\quad - \lambda C - L_2 - L_3]
\end{aligned}
\tag{1}
$$

$$
\begin{aligned}
E_1(1-x) &= yzA_5 + z(1-y)A_6 + y(1-z)A_7 + (1-y)(1-z)A_8 \\
&= yL_2 + zL_3 + [(1+\omega)I_1 - L_1 - L_2 - L_3]
\end{aligned}
\tag{2}
$$

The average expected income $\bar{E}_1$ of construction enterprises is:

$$
\bar{E}_1 = xE_1(x) + (1-x)E_1(1-x)
\tag{3}
$$

According to Formulas (1), (2), and (3), we can further obtain the replicator dynamics equation of the strategy selection of construction enterprises as follows:

$$
\begin{aligned}
F(x) &= \frac{dx}{dt} = x[E_1(x) - \bar{E}_1] = x(1-x)[E_1(x) - E_1(1-x)] \\
&= x(1-x)[yz(1+\omega)\alpha\Delta I + z(\mu+v)\lambda C - \lambda C + L_1]
\end{aligned}
\tag{4}
$$

Similarly, the expected income $E_2(y)$ or $E_2(1-y)$ of digital solution suppliers when they choose "participating in collaboration" or "not participating in collaboration" strategy are,

respectively:

$$E_2(y) = xzB_1 + z(1-x)B_2 + x(1-z)B_3 + (1-x)(1-z)B_4$$
$$= xz(1+v+\omega)\beta\Delta I + x(vI_2 + L_1) + zL_3 + [(1+\omega)I_2 - \eta C - L_1 - L_3] \tag{5}$$

$$E_2(1-y) = xzB_5 + z(1-x)B_6 + x(1-z)B_7 + (1-x)(1-z)B_8$$
$$= xL_1 + zL_3 + (I_2 - L_1 - L_3) \tag{6}$$

The average expected income $\bar{E}_2$ of digital solution suppliers is:

$$\bar{E}_2 = yE_2(y) + (1-y)E_2(1-y) \tag{7}$$

According to Formulas (5), (6), and (7), we can further obtain the replicator dynamics equation of the strategy selection of digital solution suppliers as follows:

$$F(y) = \frac{dy}{dt} = y[E_2(y) - \bar{E}_2] = y(1-y)[E_2(y) - E_2(1-y)]$$
$$= y(1-y)[xz(1+v+w)\beta\Delta I + xvI_2 + \omega I_2 - \eta C] \tag{8}$$

Similarly, the expected income $E_3(z)$ or $E_3(1-z)$ of customers when they choose "participating in collaboration" or "not participating in collaboration" strategy are, respectively:

$$E_3(z) = xyC_1 + y(1-x)C_2 + x(1-y)C_3 + (1-x)(1-y)C_4$$
$$= xy(1-\alpha-\beta)\Delta I + x(\mu+v)(1-\lambda-\eta)C + yL_2 + [I_3 -$$
$$(1-\lambda-\eta)C - L_2] \tag{9}$$

$$E_3(1-z) = xyC_5 + y(1-x)C_6 + x(1-y)C_7 + (1-x)(1-y)C_8$$
$$= 0 \tag{10}$$

The average expected income $\bar{E}_3$ of customers is:

$$\bar{E}_3 = zE_3(z) + (1-z)E_3(1-z) \tag{11}$$

According to Formulas (9), (10), and (11), we can further obtain the replicator dynamics equation of the strategy selection of customers as follows:

$$F(z) = \frac{dz}{dt} = z[E_3(z) - \bar{E}_3] = z(1-z)[E_3(z) - E_3(1-z)]$$
$$= z(1-z)\{xy(1-\alpha-\beta)\Delta I + x(\mu+v)(1-\lambda-\eta)C$$
$$+ yL_2 + [I_3 - (1-\lambda-\eta)C - L_2]\} \tag{12}$$

The three replicator dynamic equations $F(x)$, $F(y)$, and $F(z)$ can be combined to obtain a 3D dynamic system of the dynamic evolution for construction enterprises, digital solution suppliers, and customers. Let $F(x) = 0$, $F(y) = 0$, and $F(z) = 0$, the eight local equilibrium points of the dynamic system are (0,0,0), (1,0,0), (0,1,0), (0,1,0), (0,0,1), (1,1,0), (1,0,1), (0,1,1), (1,1,1), and (1,1,1), respectively, which form the equilibrium solution domain N of the three-way evolutionary game. That is, $N = \{(x, y, z)|0 \leq x \leq 1; 0 \leq y \leq 1; 0 \leq z \leq 1\}$, and there is also a mixed-strategy solution $(x^*,y^*,z^*)$ in the region. Only the pure strategy Nash equilibrium point is asymptotically stable in the three-way evolutionary game, so it is only necessary to discuss the stability of the eight pure strategy equilibrium points.

## 4.2 Stability analysis of the equilibrium point

The above equilibrium point is not completely an evolutionary stability strategy for replicating a dynamic system. It is necessary to further discuss the stability of the system equilibrium point by using the Jacobian matrix local stability analysis method proposed by Friedman [33]. The Jacobian matrix can be constructed according to the partial derivative of the three-dimensional dynamic system with respect to $x, y, z$.

$$J = \begin{pmatrix} \dfrac{\partial F(x)}{\partial x} & \dfrac{\partial F(x)}{\partial y} & \dfrac{\partial F(x)}{\partial z} \\ \dfrac{\partial F(y)}{\partial x} & \dfrac{\partial F(y)}{\partial y} & \dfrac{\partial F(y)}{\partial z} \\ \dfrac{\partial F(z)}{\partial x} & \dfrac{\partial F(z)}{\partial y} & \dfrac{\partial F(z)}{\partial z} \end{pmatrix} = \begin{pmatrix} a_{11} & a_{12} & a_{13} \\ a_{21} & a_{22} & a_{23} \\ a_{31} & a_{32} & a_{33} \end{pmatrix} \tag{13}$$

where

$$a_{11} = (1 - 2x)[L_1 - \lambda C + z(\mu + v)\lambda C + yz(1 + \omega)\alpha\Delta I]$$

$$a_{12} = -x(x - 1)z(1 + \omega)\alpha\Delta I$$

$$a_{13} = -x(x - 1)[(\mu + v)\lambda C + y(1 + \omega)\alpha\Delta I]$$

$$a_{21} = -y(y - 1)[vI_2 + z(1 + v + \omega)\beta\Delta I]$$

$$a_{22} = (1 - 2y)[\omega I_2 - \eta C + xvI_2 + xz(1 + v + \omega)\beta\Delta I]$$

$$a_{23} = -xy(y - 1)(1 + v + \omega)\beta\Delta I$$

$$a_{31} = z(z - 1)[(\mu + v)(\lambda + \eta - 1)C - y(1 - \alpha - \beta)\Delta I]$$

$$a_{32} = -z(z - 1)[L_2 + x(1 - \alpha - \beta)\Delta I]$$

$$a_{33} = (1 - 2z)[I_3 - L_2 + yL_2 + (\lambda + \eta - 1)C - x(\mu + v)(\lambda + \eta - 1)C + xy(1 - \alpha - \beta)\Delta I]$$

The eigenvalues of the Jacobian matrix corresponding to the eight equilibrium points are calculated, as shown in Table 2.

According to Lyapunov's method, the stability of the equilibrium point of the system can be judged by the sign of the eigenvalue of the Jacobian matrix. Only when all the eigenvalues of the Jacobian matrix are negative is the equilibrium point the evolutionary stability point ESS of the system [33]. In this paper, the life cycle of the digital transformation of construction enterprises is divided into three stages according to industry cycle theory: early stage, middle

**Table 2. Each equilibrium point corresponds to the eigenvalue of the Jacobian matrix.**

| Equilibrium point | $\lambda_1$ | $\lambda_2$ | $\lambda_3$ |
|---|---|---|---|
| (0,0,0) | $L_1 - \lambda C$ | $\omega I_2 - \eta C$ | $I_3 - L_2 + (\lambda + \eta - 1)C$ |
| (1,0,0) | $-(L_1 - \lambda C)$ | $(\omega + v)I_2 - \eta C$ | $I_3 - L_2 + (1 - \mu - v)(\lambda + \eta - 1)C$ |
| (0,1,0) | $L_1 - \lambda C$ | $-(\omega I_2 - \eta C)$ | $I_3 + (\lambda + \eta - 1)C$ |
| (0,0,1) | $L_1 - (1 - \mu - v)\lambda C$ | $\omega I_2 - \eta C$ | $-[I_3 - L_2 + (\lambda + \eta - 1)C]$ |
| (1,1,0) | $-(L_1 - \lambda C)$ | $-[(\omega + v)I_2 - \eta C]$ | $I_3 + (1 - \mu - v)(\lambda + \eta - 1)C + (1 - \alpha - \beta)\Delta I$ |
| (1,0,1) | $-[L_1 - (1 - \mu - v)\lambda C]$ | $(\omega + v)I_2 - \eta C + (1 + v + \omega)\beta\Delta I$ | $-[I_3 - L_2 + (1 - \mu - v)(\lambda + \eta - 1)C]$ |
| (0,1,1) | $L_1 - (1 - \mu - v)\lambda C + (1 + \omega)\alpha\Delta I$ | $-(\omega I_2 - \eta C)$ | $-[I_3 + (\lambda + \eta - 1)C]$ |
| (1,1,1) | $-[L_1 - (1 - \mu - v)\lambda C + (1 + \omega)\alpha\Delta I]$ | $-[(\omega + v)I_2 - \eta C + (1 + v + \omega)\beta\Delta I]$ | $-[I_3 + (1 - \mu - v)(\lambda + \eta - 1)C + (1 - \alpha - \beta)\Delta I]$ |
| $(x^*, y^*, z^*)$ | $\lambda_1^*$ | $\lambda_2^*$ | $\lambda_3^*$ |

**Table 3. Each equilibrium point corresponds to the eigenvalue symbol judgment.**

| Equilibrium point | Case 1 | Case 2 | Case 3 |
|:---:|:---:|:---:|:---:|
| (0,0,0) | −−− | −−− | +++ |
| (1,0,0) | +−− | +++ | −++ |
| (0,1,0) | −+⊗ | −+⊗ | +−+ |
| (0,0,1) | −−+ | +−+ | ++− |
| (1,1,0) | ++⊗ | +−+ | −−+ |
| (1,0,1) | +−+ | −+− | −+− |
| (0,1,1) | ⊗+⊗ | ++⊗ | +−− |
| (1,1,1) | ⊗+⊗ | −−− | −−− |

stage, and mature stage. The collection and analysis of data and the research and development of digital technology are the basis for the digital transformation of enterprises. Enterprises need to invest in digital technology, which is the main task in the early stage of digital transformation. With the accumulation of data, the value of a large amount of data needs to be developed. By building a digital platform to fully activate data and starting to establish contacts with other companies in data sharing to develop value networks, services across company boundaries will be provided to create value for customers, and digital transformation will enter the middle stage [34]. The continuous expansion of customer groups will lead to a diversity of customer needs, which will be met by various digital solutions. Digital services will enter the stage of mass customization, and digital transformation will enter the mature stage [35]. Combined with the analysis of the above stages and the Jacobian matrix eigenvalue expression corresponding to each equilibrium point in Table 2, the stability analysis of each equilibrium point is carried out under the following three conditions, and the specific analysis results are shown in Table 3.

Case 1: Construction enterprises are in the early stage of digital transformation, and the level of digitalization is low. They need to invest considerable manpower, material resources, time, and capital to learn and absorb digital technology and knowledge. Therefore, the cost of implementing the whole value chain collaborative supervision for construction enterprises is high, and the digital profits and benefits are not significant. When $(1 - \mu - v)\lambda C > L_1, (v + \omega)I_2 + (1 + v + \omega)\beta\Delta I < \eta C, I_3 + (1 - \alpha - \beta)\Delta I < (1 - \mu - v)(1 - \lambda - \eta)C + L_2$, it is found that the equilibrium point of evolutionary stability can only be (0,0,0). In the early stage of digital transformation, for construction enterprises, digital solution suppliers, and customers, when at least one party chooses collaborative supervision or to collaboration, the coordination costs paid by the entities involved in value co-creation are greater than the benefits obtained. To achieve their own goal of maximizing benefits, the three parties will give up the cooperation strategy with the growth of the number of evolution periods, and the value co-creation strategy of the three parties will finally evolve into no collaborative supervision, no collaboration, or no collaboration, respectively.

Case 2: When the digital transformation of construction enterprises has been ongoing for a period of time, the enterprises have initially mastered a certain amount of digital resources and technologies. The digital level is in the middle, the cost of collaborative supervision is reduced, and the digital level coefficient is increased. In this case, although $\lambda C > L_1, (1 - \mu - v)\lambda C < L_1, (v + \omega)I_2 > \eta C > \omega I_2, (1 - \lambda - \eta)C + L_2 > I_3 > (1 - \mu - v)(1 - \lambda - \eta)C + L_2$, two equilibrium points (0,0,0) and (1,1,1) can be identified to achieve steady equilibrium. It shows that with the growth of evolution periods, game players will eventually choose to participate in value co-creation or not to achieve the goal of maximizing benefits, and the system will eventually evolve into a state of complete non-collaboration or complete collaboration. Compared

with Case 1, it is found that the improvement of the digital level encourages game players to choose the strategy of participating in value co-creation.

Case 3: The digital transformation of construction enterprises has entered a mature stage. Enterprises already have rich digital technologies and resources, and the construction of digital platforms is relatively mature. The digital level is higher, the cost of collaborative supervision is lower, and the coefficient of the digital level is higher. At the same time, the importance of customers in the entire value chain has also increased dramatically. In this case, $\lambda C < L_1, \omega I_2 > \eta C, (1 - \mu - v)(1 - \lambda - \eta)C + L_2 < I_3$; thus, it can be concluded that the equilibrium point of the system evolutionary stability strategy is only (1,1,1). In the mature stage of digital transformation, the benefits gained by all three parties participating in value creation are greater than the collaborative costs paid. With the continuous growth of the evolution period, the strategy of game players participating in value co-creation in the process of maximizing profits will eventually evolve into a fully collaborative state, i.e., collaborative supervision, collaboration, and collaboration, respectively. On the other hand, it also shows that when the digitalization level of enterprises is very high, the choice of collaborative innovation is more conducive to maximizing the interests of enterprises and high-quality development.

## 5. Simulation analysis

### 5.1 Evolutionary path of tripartite participants at different stages

To verify the validity of evolutionary stability analysis and more intuitively describe the dynamic evolution of the collaborative strategy of value co-creators in the early stage, the middle stage, and the mature stage during the service-oriented digital transformation of the construction industry, this paper conducts a numerical simulation of the optimal evolutionary stability strategies in the above three stages. At the same time, according to the suggestions of relevant experts and with reference to the analog values of relevant parameters set by Gao [36], Peng [37], and Mei [38] *et al.*, the parameters in the early, middle and mature stages of digital transformation were assigned.

In the early stage of digital transformation, the initial values of each parameter are $I_2 = 20$, $I_3 = 10, \alpha = \beta = 1/3, \Delta I = 30, L_1 = 4, L_2 = 6, C = 60, \lambda = \eta = 1/3$, and $\mu = v = \omega = 0.1$. The above parameters meet the conditions of Case 1 and evolve 50 times from different initial strategy combinations. The evolution results are shown in Fig 2. The results show that the system eventually evolves into a stable strategy combination, i.e., no collaborative supervision, no collaboration, and no collaboration, respectively, under different combinations of the initial strategy probability values of the game players, which is consistent with the conclusion of Case 1. It shows that even though there is a certain proportion of construction enterprises, digital solution suppliers and customers that initially choose to participate in value co-creation, the low level of digitalization makes the cooperation costs between construction enterprises and digital solution suppliers higher, the channels for customers to collaborate are fewer, and the three parties do not participate in value co-creation to obtain greater benefits. Driven by interests, the three parties eventually tend to choose not to collaborate.

In the middle stage of digital transformation, the initial values of each parameter are $I_2 = 20, I_3 = 10, \alpha = \beta = 1/3, \Delta I = 40, L_1 = 7, L_2 = 6, C = 30, \lambda = \eta = 1/3$, and $\mu = v = \omega = 0.4$. The above parameters meet the conditions of Case 2 and evolve 50 times from different initial strategy combinations. The evolution results are shown in Fig 3. The results show that there are two stable strategies under different combinations of the initial strategy probability values of the game players, i.e., all three parties collaborate or do not collaborate, which is consistent with the conclusion of case 2. It shows that the construction capacity and service efficiency of

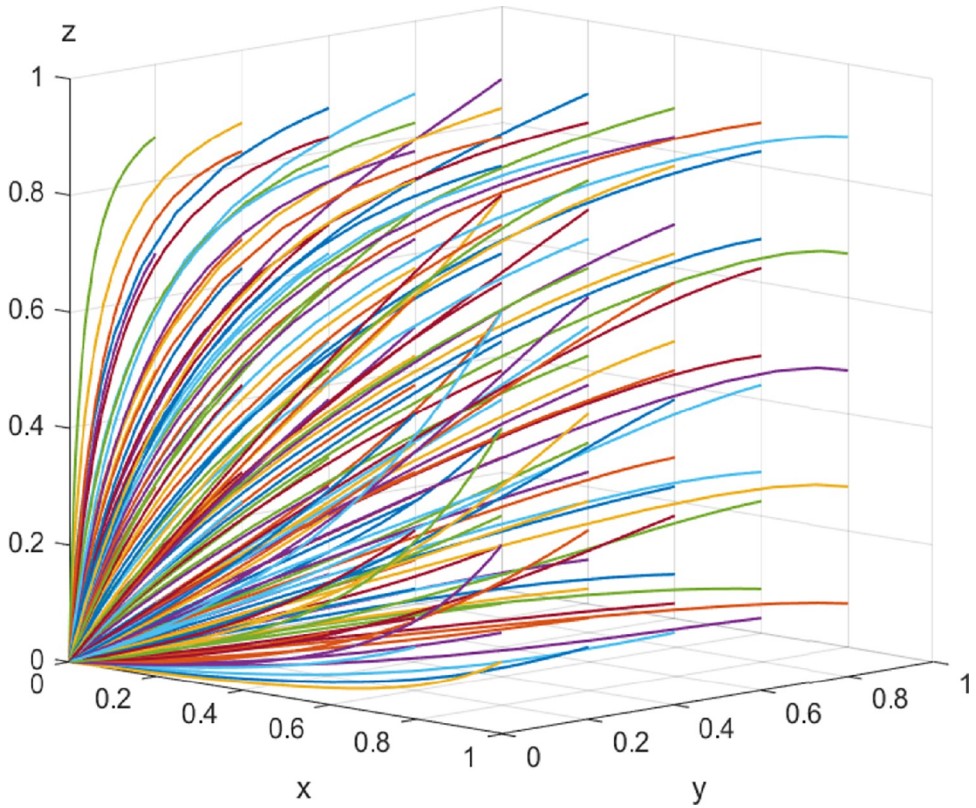

**Fig 2. Evolutionary paths in the early stage of digital transformation.**

enterprises have improved with the development of digitalization, thus, increasing the total revenue of the three parties. At the same time, the application of digital technology provides a variety of collaboration channels, which promotes effective communication and knowledge sharing among various entities, reduces their collaboration costs, and greatly increases the possibility of choosing to participate in collaborative innovation.

In the mature stage of digital transformation, the initial values of each parameter are $I_2 = 20$, $I_3 = 10$, $\alpha = \beta = 1/3$, $\Delta I = 50$, $L_1 = 7$, $L_2 = 7$, $C = 20$, $\lambda = \eta = 1/3$, and $\mu = v = \omega = 0.85$. The above parameters meet the conditions of Case 3 and evolve 50 times from different initial strategy combinations. The evolution results are shown in Fig 4. The results show that the system eventually evolves into a stable strategy,i.e., collaborative supervision, collaboration, and collaboration, respectively, under different combinations of the initial strategy probability values of the game players, which is consistent with the conclusion of Case 3. It shows that even though there is a certain proportion of construction enterprises, digital solution suppliers, and customers who choose not to collaborate at first, high-level enterprise digitalization can bring high benefits and low costs to all entities. Compared with the medium term, the loss of the three parties not cooperating will increase. Driven by interests, the three parties eventually tend to choose to participate in cooperation.

According to the constructed Jacobi matrix, it can be seen that the equilibrium point is affected by different parameters of different subjects, which makes the three parties form different equilibrium states in different stages of Digital transformation. However, according to the above simulation analysis, from the perspective of the whole Digital transformation cycle, the strategy combination (0, 0, 0) in the early stage will develop into the strategy combination

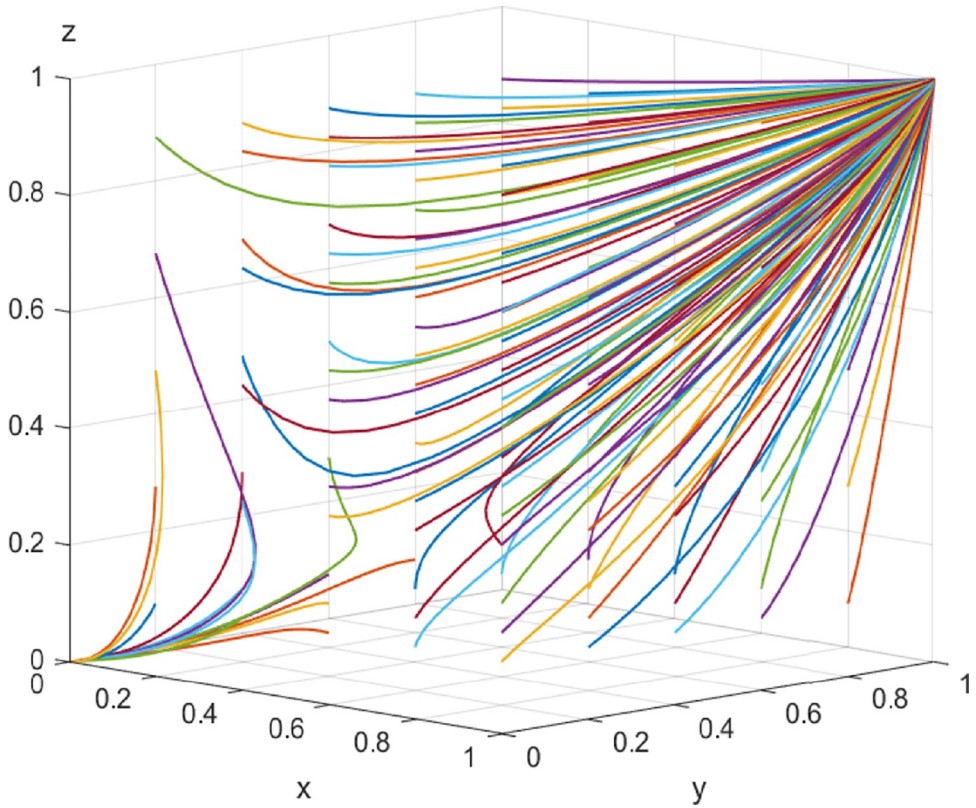

**Fig 3. Evolutionary paths in the middle stage of digital transformation.**

(0, 0, 0) or (1, 1, 1) in the middle stage, and finally will reach the equilibrium state of (1, 1, 1) in the mature stage. It indicates that with the improvement of the digitalization level, the value co-creation entities have finally reached the ideal state of (construction enterprises cooperating in supervision, digital solution suppliers participating in cooperation, customers participating in cooperation), and H1 and H4 are accepted. At the same time, the strategy of digital solution suppliers changes with the strategy of construction enterprises, and the strategic choice of customers is also synchronized with digital solution suppliers, H2 and H3 are accepted. Goldfar and Tucker [39] believe that digital technology has achieved permanent and real-time data sharing, fully reduced the degree of information asymmetry in enterprises, and thus reduced the cost of collaborative innovation between enterprises and customers. At the same time, digital transformation helps enterprises shorten the cycle of product production and innovation, further amplifying the driving effect of interests [40]. Obviously, the higher the level of digitalization, the greater the difference between collaborative benefits and costs, and the system will inevitably evolve into an ideal state of complete collaboration eventually.

## 5.2 Simulation of key parameters

There are two distinct stability strategies in the middle stage of digital transformation, and the entire system will eventually develop into a fully collaborative state in the mature stage. Therefore, it is necessary to further study the impact of the main parameters in the middle stage on the evolution of game players' behavior to promote the service-oriented digital transformation of the construction industry to achieve the ideal state of full collaboration among value co-creators faster.

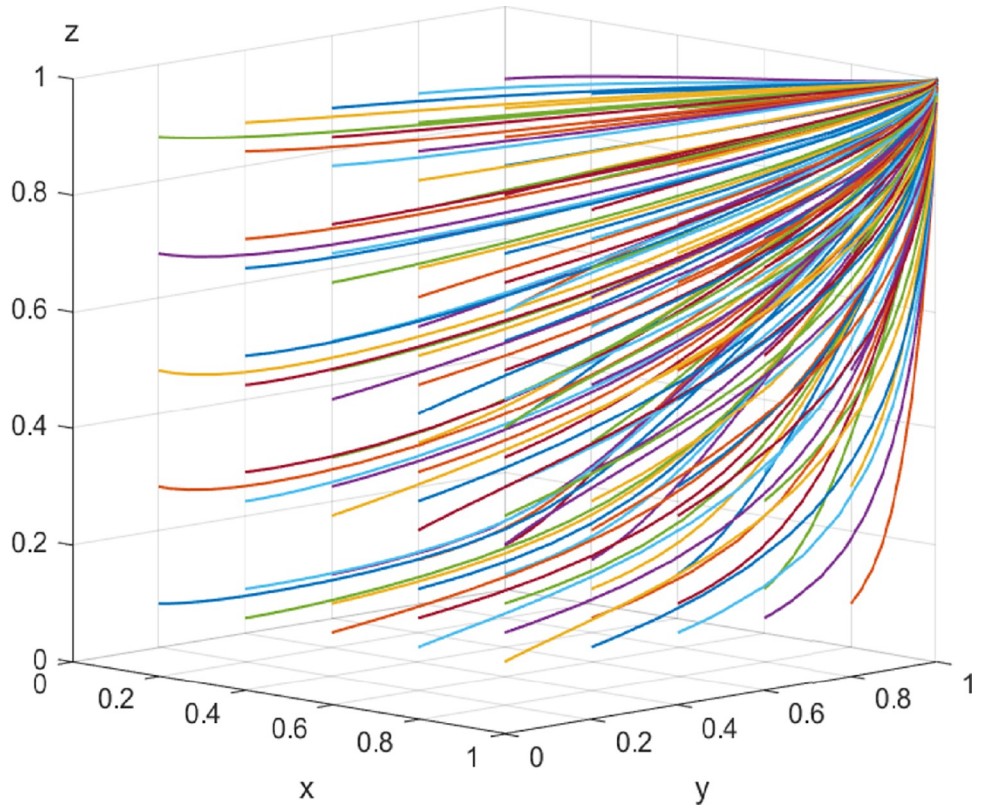

**Fig 4. Evolutionary paths in the mature stage of digital transformation.**

(1) Influence of initial willingness to collaborate on the system

On the basis of the values taken in the middle stage of digital transformation, other variables are assumed to remain unchanged, and the initial values of $x, y$, and $z$ are taken as variables. We set $x = y = z = 0.2$, $x = y = z = 0.3$, and $x = y = z = 0.4$, and the impact of the initial willingness to cooperate of game players on their decision-making behavior is shown in Fig 5. It can be seen from the figure that the initial willingness to cooperate has an impact on the evolution of the behavior strategies of the game players in the middle stage of digital transformation. With the increase in the initial willingness to cooperate, the system gradually evolves from the stable equilibrium point (0,0,0) of complete non-collaboration to the stable equilibrium point (1,1,1) of complete collaboration, and the critical value of the initial collaborative willingness leading to this change is between 0.2 and 0.3. After exceeding the critical value, for the whole system, the higher the initial willingness of the three parties to cooperate, the faster the system will eventually evolve into a stable state of complete cooperation. For a single agent, digital solution suppliers always reach a stable state at a fast speed and choose to collaborate.

(2) Influence of the digital level coefficient on the system

On the basis of the values taken in the middle stage of digital transformation, the impact of the initial willingness to cooperate of the game players on the system evolution results is eliminated, so the probability of the initial collaboration strategy of construction enterprises, digital solution suppliers, and customers is 0.2. We assume that other variables remain unchanged and set $\mu = 0.3$, $\mu = 0.4$, and $\mu = 0.7$. The simulation results are shown in Fig 6. With the increase in data collection and analysis coefficients, the evolution results of the system have

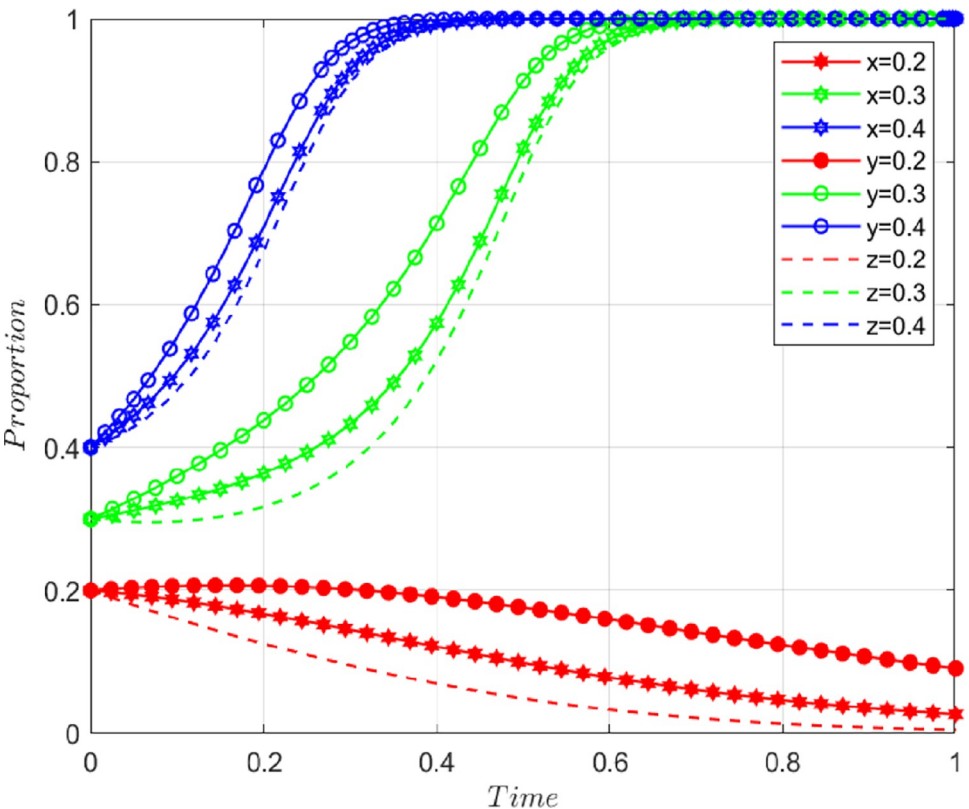

**Fig 5. Effect of initial willingness to cooperate on the evolutionary path.**

not changed significantly. When the initial collaborative willingness of the game players is low, the changes in data collection and analysis coefficient will not change the final strategy of the game players in the middle stage of the digital transformation.

We assume that other variables remain unchanged and set $v = 0.3$, $v = 0.4$, and $v = 0.7$. The simulation results are shown in Fig 7. With the increase in the platform coefficient, the probability of digital solution suppliers collaborating increases, while the probability of construction enterprises' collaborative supervision decreases. Although the evolutionary process of the system has changed, the evolutionary stability strategy of the game players is still no collaborative supervision, no participation in collaboration, and no participation in collaboration, respectively. In other words, when the initial collaborative willingness of game players is low, the change in the platform coefficient will not affect the evolutionary stability strategy combination of the system.

We assume that other variables remain unchanged and set $\omega = 0.3$, $\omega = 0.4$, and $\omega = 0.7$. The simulation results are shown in Fig 8. With the increase in digital coefficients in the construction process, the evolutionarily stable equilibrium point of the system gradually tends to (1,1,1) from (0,0,0). It shows that the value co-creation behavior of game players will gradually evolve from a completely uncooperative stable equilibrium point (0,0,0) to a completely cooperative stable equilibrium point (1,1,1) with the increase in the digital coefficient in the construction process even if the initial willingness to cooperate of game players is lower than the critical value. The reason is that the digitalization of the construction process optimizes the business chain and supply chain of construction enterprises and digital solution suppliers in an all-around way, and the digitalization business runs through all the activities of the whole life cycle of product design, production, construction, service, and so on. It makes the business

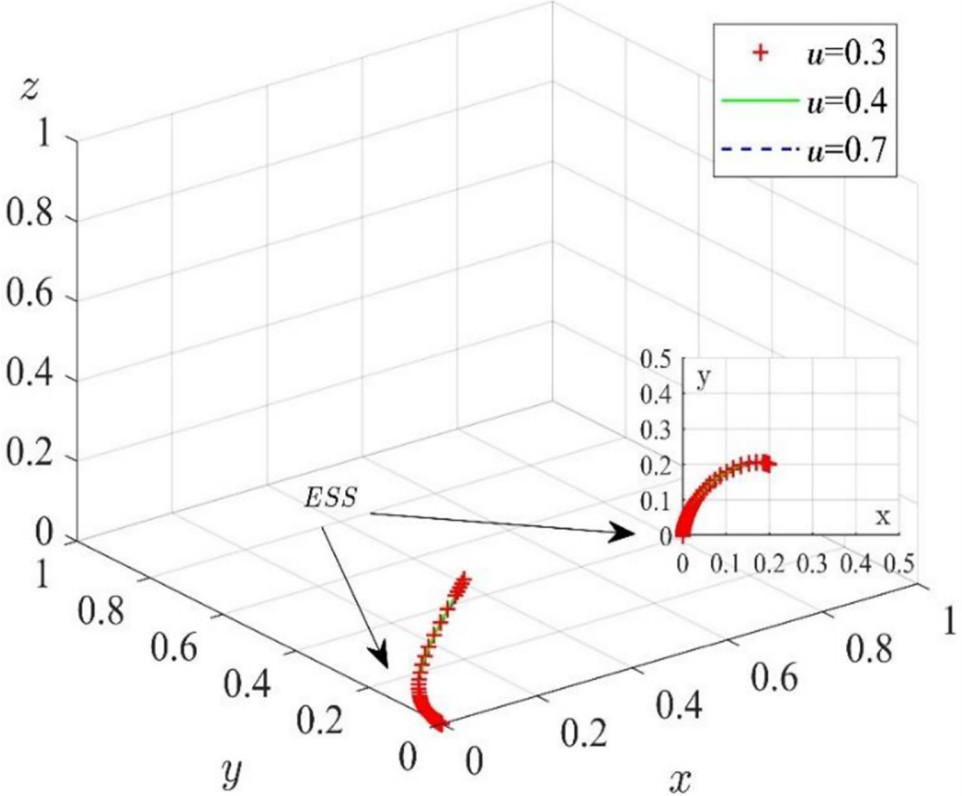

**Fig 6. Effect of data collection and analysis coefficient on the evolutionary path.**

activities of both parties more transparent and easier to communicate and reduces the supervision costs of construction enterprises and the collaboration costs of both enterprises. At the same time, the digitalization of the construction process also provides a channel for customers to participate in activities of the entire the whole life cycle, reducing their participation costs and increasing their value perception. The profits obtained by the three parties participating in the collaboration are far higher than the cost they have to pay. The motivation of pursuing profit maximization urges them to choose to participate in the collaboration.

Furthermore, by comparing the results of Figs 6–8, it is found that the digitalization level of the construction process is the key factor in promoting the development from the middle stage of the digitalization transformation to the mature stage and is an important catalyst and breakthrough for promoting the service-oriented digitalization transformation of the construction industry. This conclusion is consistent with the views of other scholars. The collection and analysis of data only provide a framework for the composition of related technologies, and the digital platform provides new channels for the cooperation of value co-creators. However, the embodiment of their value still depends on production activities. Compared with the level of data collection and analysis and platformization, the digitalization of the construction process has a more direct effect on the collaborative intention of value co-creators [41]. The application of digital technology in the construction process is the focus of construction enterprises. The leaders of government and enterprises should speed up the improvement of the digital infrastructure of construction enterprises, promote the integration and application of advanced digital construction technology in the construction process, and optimize the interaction and construction capacity of various departments.

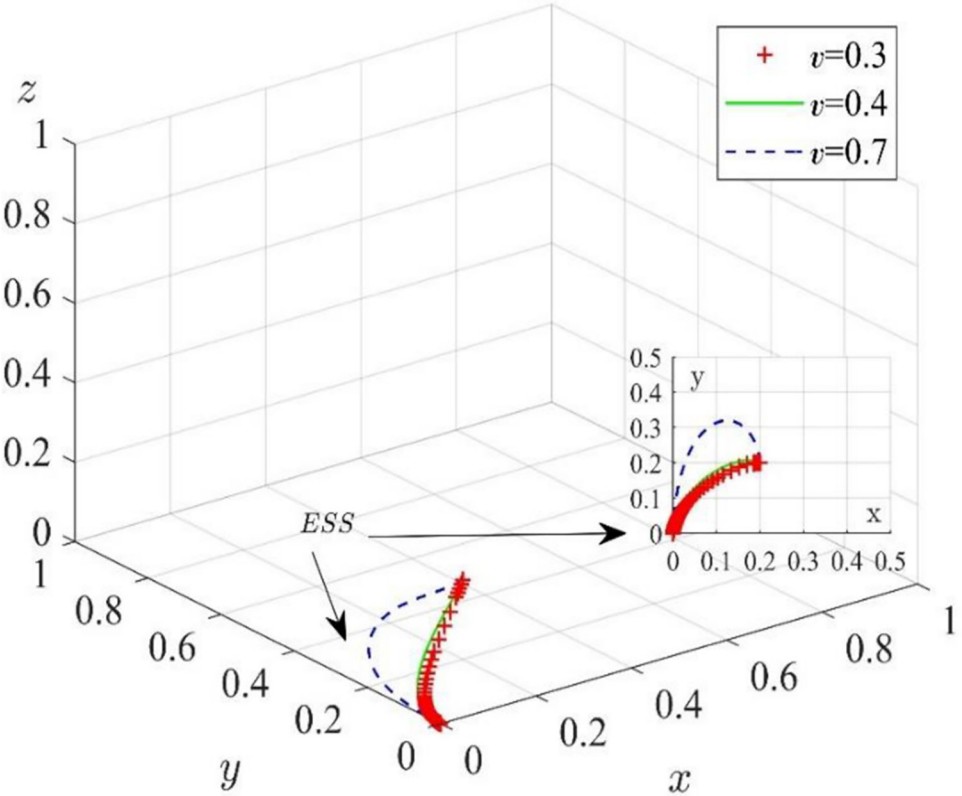

**Fig 7. Effect of the platform coefficient on the evolutionary path.**

## 6. Conclusion

In this paper, a tripartite evolutionary game model composed of construction enterprises, digital solution suppliers, and customers is constructed, aiming at the problem of uncertain strategy combinations generated by multiple agents in pursuit of the maximization of their own interests in the service-oriented digital transformation process of the construction industry. We reveal the collaborative evolutionary path of various value-creating agents in the process of digital transformation and the impact of relevant elements on the realization mechanism of value co-creation. According to the influence relationship and stability conditions of various factors, we propose relevant countermeasures and suggestions for the digital transformation and upgrading of the construction industry. The main conclusions of the study are as follows:

First, the level of digitalization has a direct impact on the synergistic benefits and costs of construction enterprises, digital solution suppliers, and customers. At different stages of digital transformation, the combination of behavior strategies of game players will change. The higher the level of digitalization is, the higher the degree of collaboration among the value co-creators is. Value co-creators in the service-oriented digital ecosystem of the construction industry will eventually choose a stable combination of completely collaborative strategies. The collaborative behavior of the subjects has an impact on the realization of their own interests maximization and the value creation efficiency of the value chain. Second, in the middle stage of the digital transformation, the initial willingness to cooperate of the game players will eventually make the system evolve into two types of completely opposite stable strategies, and the higher the initial willingness to cooperate of the players is, the faster they will develop to the complete collaboration state in the mature stage. Finally, in the middle stage of the digital

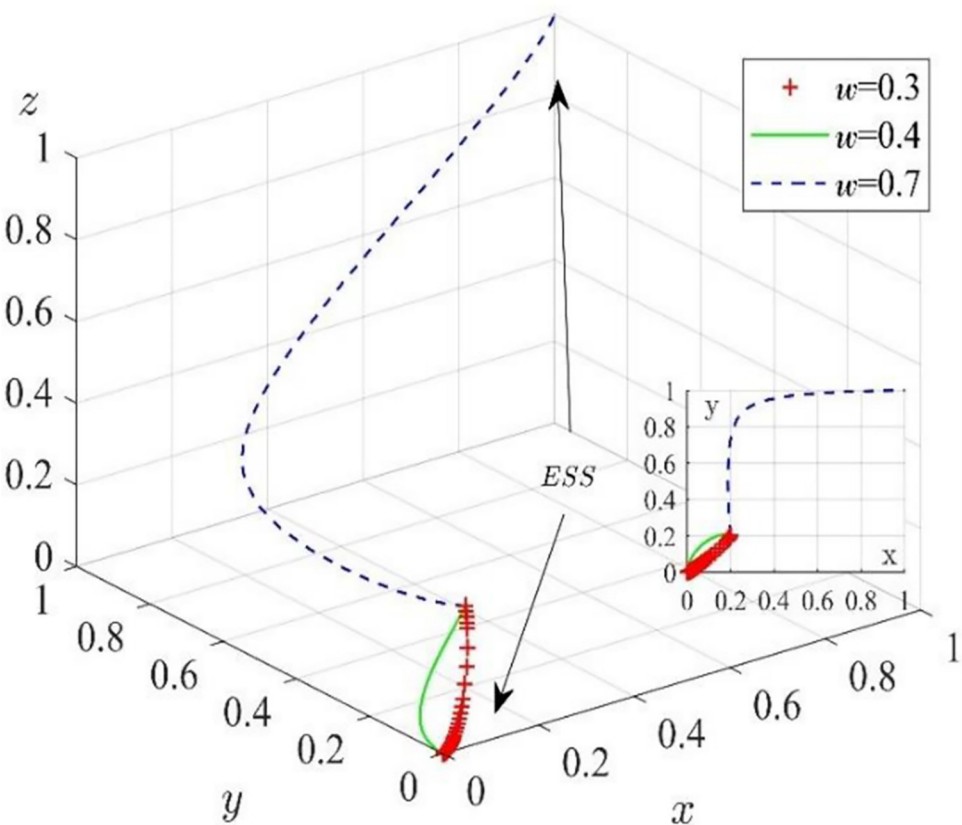

**Fig 8. Effect of digitization coefficient of the construction process on the evolutionary path.**

transformation, when the initial willingness to cooperate of the three parties is lower than the critical value, the improvement of the digital level of the construction process can reverse the final evolution of the system into a completely uncooperative state, which is the key to promoting the development of the system and the cooperation of the agents.

To promote cooperation among value co-creators and achieve the balance of the service-oriented digital ecosystem of the construction industry, the following suggestions are proposed:

(1) At different stages of digital transformation, construction enterprises should adjust the focus of collaborative development according to the level of digital transformation and the actual situation of each participant to reduce the participation cost of value co-creation. Specifically, in the early stage of digital transformation, the development of digital technology is the focus of the construction industry. Construction enterprises should pay attention to the improvement of digital infrastructure, build a platform together with digital solution suppliers to provide customers with channels to participate in value co-creation, and reduce the difficulty of tripartite participation. In the middle stage of digital transformation, construction enterprises should pay attention to the improvement of the digital level in the construction process, promote the integration and application of advanced digital construction technology in the construction process, and promote the transition of the industry to the mature stage of digital transformation. In the mature stage of digital transformation, the optimization of the income distribution mechanism should be considered, and cooperation

and trust among participants should be enhanced to maintain the stability of the value co-creation network.

(2) Change the concept of all participants in the service-oriented value chain of the construction industry and establish a win-win attitude. First, construction enterprises, digital solution suppliers, and customers themselves need to change their mindset and regard each other as partners for value co-creation. Second, the government should give full play to its guiding role, actively use the new national system, and fully mobilize the collaborative innovation and willingness of value co-creators.

Although this paper has conducted a useful discussion on the collaborative behavior of value co-creators in the digital service ecosystem of the construction industry, there are still some limitations. Firstly, in terms of research methods, due to the difficulty in obtaining data related to the digital transformation of service-oriented construction enterprises, this paper carries out research based on artificially simulated data. In future research, a specific construction enterprise should be taken as an example and real data should be used for more convincing simulation analysis; Secondly, in terms of research content, this article only considers the behavior strategies of three value co-creators, namely construction enterprises, digital solution suppliers, and customers. However, under the special national conditions of China, the government plays a key role in the transformation process of enterprise digitization and service. In the future, it is possible to further explore the four-party evolutionary game considering the government under the national conditions of socialism with Chinese characteristics.

## Supporting information

**S1 File.**
(ZIP)

## Author Contributions

**Conceptualization:** Shiming Wang.

**Data curation:** Hui Su.

**Formal analysis:** Hui Su.

**Funding acquisition:** Shiming Wang.

**Investigation:** Hui Su.

**Methodology:** Hui Su, Qiang Hou.

**Project administration:** Shiming Wang.

**Resources:** Hui Su.

**Software:** Hui Su.

**Supervision:** Shiming Wang.

**Validation:** Qiang Hou.

**Visualization:** Hui Su.

**Writing – original draft:** Hui Su.

**Writing – review & editing:** Shiming Wang, Qiang Hou.

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
