## [Decision Letter · Decision Letter 0]

20 Feb 2023

PONE-D-23-02979Evolutionary game study on multi-agent value co-creation of service-oriented digital transformation in the construction industryPLOS ONE

Dear Dr. Su,

Thank you for submitting your manuscript to PLOS ONE. After careful consideration, we feel that it has merit but does not fully meet PLOS ONE’s publication criteria as it currently stands. Therefore, we invite you to submit a revised version of the manuscript that addresses the points raised during the review process.

Please submit your revised manuscript by Apr 06 2023 11:59PM. If you will need more time than this to complete your revisions, please reply to this message or contact the journal office at plosone@plos.org. Please include the following items when submitting your revised manuscript:A rebuttal letter that responds to each point raised by the academic editor and reviewer(s). You should upload this letter as a separate file labeled 'Response to Reviewers'.A marked-up copy of your manuscript that highlights changes made to the original version. You should upload this as a separate file labeled 'Revised Manuscript with Track Changes'.An unmarked version of your revised paper without tracked changes. You should upload this as a separate file labeled 'Manuscript'.

We look forward to receiving your revised manuscript.

Kind regards,

Simon Grima, PhD

Academic Editor

PLOS ONE

Journal Requirements:

"This work was supported in part by the Basic Scientific Research Project of Education Department of Liaoning Province under grant LJKMR20220709 and in part by the Social Science Fund of Liaoning Province under grant L21BGL027."

Reviewers' comments:

Reviewer's Responses to Questions

**Comments to the Author**

1. Is the manuscript technically sound, and do the data support the conclusions?

Reviewer #1: Yes

Reviewer #2: Partly

2. Has the statistical analysis been performed appropriately and rigorously? 

Reviewer #1: Yes

Reviewer #2: N/A

3. Have the authors made all data underlying the findings in their manuscript fully available?

Reviewer #1: Yes

Reviewer #2: Yes

4. Is the manuscript presented in an intelligible fashion and written in standard English?

Reviewer #1: Yes

Reviewer #2: Yes

5. Review Comments to the Author

Reviewer #1: The paper addresses an interesting topic related to the service-oriented digital transformation of the construction industry. I acknowledge the amount of work invested in preparing the manuscript and, overall, I believe it strengthens the literature in this scientific field. At the same time, however, I consider that there are several sources of improvement, that need to be addressed. I would recommend that the author(s) reconsider and improve the following:

- In the Introduction, the author(s) should clearly state how the research performed detaches from other studies since the topic is extensively approached in the literature, but this research views it from a different angle: why is it different from previous research, justify it, please. Also, please clearly state/underline the innovations brought by this research in the scientific field, along with the authors’ own extensive contribution; the introduction should comprise a final paragraph about the structure of the paper to orient the reader about the following sections.

- Please further substantiate the theoretical framework and provide additional groundings to support the work hypotheses; the hypotheses should be followed throughout the paper/research design with clear mentions about their validation/ rejection according to the results obtained.

- I would suggest further outlining the importance of the results obtained. Please explain in more detail the results and relate them to other empirical findings and theoretical grounds.

- Other sections which would benefit from further work would be the policy and managerial implications of own findings, along with research limitations (including specific measures to cope with these limitations) and future research directions.

Overall, I consider that the paper would contribute to the literature, but some more attention and discussion are needed.

Reviewer #2: 1. The abstract requires being built upon a standardised method of compiling, ie; aim, method, result, concluding framework. It is not clear how the ultimate aim is connected to the application for a need of construction processes - there is no discussion about this in the beginning of the literature chapter. ie: why is game theory necessary to construction? Is there reference made to current digitisation of construction building processes as has been carried out to this day presented in any part herewith?

2. The literature chapter will benefit from added discussion in relation to the opening sections of the paper by adding on the changes of the last 10 years in terms of digitisation of construction; there is no mention how the need of game theory has evolved to satisfy a demand. What is the demand?

3. Statistical analysis is not being questioned here

4. Kindly revisit the global discourse on the digitisation strategies that exist between customers and construction stakeholders; for eg: 'Realizing the Need for Digital Transformation of Stakeholder Management: A Systematic Review in the Construction Industry', 2021.

5. Kindly give this paper a context since it seems to be divorced from the reality of need in the sector.

6. PLOS authors have the option to publish the peer review history of their article (what does this mean?). If published, this will include your full peer review and any attached files.

Reviewer #1: No

Reviewer #2: No

<quillbot-extension-portal></quillbot-extension-portal>

---

## [Author Response · Author response to Decision Letter 0]

12 Apr 2023

Reviewer #1’s Comments: The paper addresses an interesting topic related to the service-oriented digital transformation of the construction industry. I acknowledge the amount of work invested in preparing the manuscript and, overall, I believe it strengthens the literature in this scientific field. At the same time, however, I consider that there are several sources of improvement, that need to be addressed. I would recommend that the author(s) reconsider and improve the following:

1.In the Introduction, the author(s) should clearly state how the research performed detaches from other studies since the topic is extensively approached in the literature, but this research views it from a different angle: why is it different from previous research, justify it, please. Also, please clearly state/underline the innovations brought by this research in the scientific field, along with the authors’ own extensive contribution; the introduction should comprise a final paragraph about the structure of the paper to orient the reader about the following sections.

2.Please further substantiate the theoretical framework and provide additional groundings to support the work hypotheses; the hypotheses should be followed throughout the paper/research design with clear mentions about their validation/ rejection according to the results obtained.

3.I would suggest further outlining the importance of the results obtained. Please explain in more detail the results and relate them to other empirical findings and theoretical grounds.

4.Other sections which would benefit from further work would be the policy and managerial implications of own findings, along with research limitations (including specific measures to cope with these limitations) and future research directions.

Response: We sincerely wish to thank the reviewers for his/her valuable and constructive comments which are very helpful in improving our manuscript.

1.In the revision, the introduction has been re-edited, we further discuss the differences between this study and previous studies based on the sorting of existing literature and indirectly explain the innovation of this article based on it. Specifically, previous research on the value co-creation of service ecosystems in the context of digitalization has mostly focused on the manufacturing industry, while research on the integration of digitalization and services in the construction industry mainly focuses on promoting the service efficiency of construction enterprises and the entire service ecosystem through digitalization. This article takes the digital service ecosystem of the construction industry as the research object, focusing on the collaborative behavior among value co-creators in the value chain, and incorporating digital-level factors that have not been addressed in existing research into the research scope, to preliminary explore the integration of digital and service-oriented in the construction industry (Line 91-94, page 4 and Line 193-199, page 9). In addition, we have added a discussion on research contributions (Line 95-104, page 4) and introduced the main structure of this article in the last paragraph of the introduction (Line 105-113, page 5). 

2.In the revised paper, we will place the analysis section of the collaboration mechanism among game entities in Section 3.2 and propose corresponding theoretical assumptions (see Section 3.2 in red font). For example, 

H1: Under the supervision of construction enterprises, customers will ultimately choose to participate in the collaboration.

H2: Whether digital solution suppliers participate in collaborative innovation depends on the strategic choices of construction enterprises.

H3: The customers’ strategic choice is synchronized with the digital solution suppliers.

H4: The digital level can promote the collaborative value co-creation of game players.

Based on this, we will modify the title of Section 3.2 to “Analysis and Research Assumptions of Tripartite Games”, and then explicitly mention the acceptance/rejection assumption in the results discussion section based on the simulation results (Line 482-498, page 25). The assumptions and descriptions of relevant parameters for constructing the game model in section 3.2 of the old version have been included in section 3.3 of the new version (see section 3.3 in red font).

3.In the revision, In the analysis section of the simulation results, we discuss the connection between our results and other empirical findings and provided relevant literature support. Specifically, the analysis results of the evolution trend of collaborative behavior among value co-creators supplement the following discussion (Line 492-498, page 25):

Goldfar and Tucker [37] believe that digital technology has achieved permanent and real-time data sharing, fully reduced the degree of information asymmetry in enterprises, and thus reduced the cost of collaborative innovation between enterprises and customers. At the same time, digital transformation helps enterprises shorten the cycle of product production and innovation, further amplifying the driving effect of interests [38]. Obviously, the higher the level of digitalization, the greater the difference between collaborative benefits and costs, and the system will inevitably evolve into an ideal state of complete collaboration eventually.

The analysis of the impact of key parameters on the evolution results is supplemented by the following discussion (Line 558-563, page 28)：

This conclusion is consistent with the views of other scholars. The collection and analysis of data only provide a framework for the composition of related technologies, and the digital platform provides new channels for the cooperation of value co-creators. However, the embodiment of their value still depends on production activities. Compared with the level of data collection and analysis and platformization, the digitalization of the construction process has a more direct effect on the collaborative intention of value co-creators [39]. 

4.In the new version, we re-edit the conclusions, discuss the limitations of the research, and provide specific countermeasures. Firstly, in terms of research methods, due to the difficulty in obtaining data related to the digital transformation of service-oriented construction enterprises, this paper carries out research based on artificially simulated data. In future research, a specific construction enterprise should be taken as an example and real data should be used for more convincing simulation analysis; Secondly, in terms of research content, this article only considers the behavior strategies of three value co-creators, namely construction enterprises, digital solution suppliers, and customers. However, under the special national conditions of China, the government plays a key role in the transformation process of enterprise digitization and service. In the future, it is possible to further explore the four-party evolutionary game considering the government under the national conditions of socialism with Chinese characteristics (see section 6 in red font).

Reviewer #2’s Comments:

1.The abstract requires being built upon a standardised method of compiling, ie; aim, method, result, concluding framework. It is not clear how the ultimate aim is connected to the application for a need of construction processes - there is no discussion about this in the beginning of the literature chapter. ie: why is game theory necessary to construction? Is there reference made to current digitisation of construction building processes as has been carried out to this day presented in any part herewith?

2.The literature chapter will benefit from added discussion in relation to the opening sections of the paper by adding on the changes of the last 10 years in terms of digitisation of construction; there is no mention how the need of game theory has evolved to satisfy a demand. What is the demand?

3.Kindly revisit the global discourse on the digitisation strategies that exist between customers and construction stakeholders; for eg: 'Realizing the Need for Digital Transformation of Stakeholder Management: A Systematic Review in the Construction Industry', 2021.

4.Kindly give this paper a context since it seems to be divorced from the reality of need in the sector.

Response: The authors would like to gratefully thank the reviewer for his/her valuable and constructive comments.

1.In the revision, we have re-edited the abstract based on the framework of objectives, methods, results, and conclusions, and added a discussion on the research objectives and methods (Line 43-48, page 2). In the literature review section, we have supplemented the connection between our research objectives and the needs of the construction process (Line 182-187, page 8). At the same time, by citing previous relevant literature, It is necessary to verify the game theory method for this study (Line 131-132, page 6; Line 145-148, page 6-7). In addition, a discussion on the digitization of the ongoing construction process was added (Line 169-176, page 8).

2.In the literature review section, we further discussed the application of relevant digital technologies in construction processes, indicating the changes in building digitization over the past decade (Line 169-176, page 8). In addition, it proves that game theory is applicable to this study and necessary (Line 131-132, page 6; Line 145-148, page 6-7).

3.In the new version, we focused on the literature “Realizing the Need for Digital Transformation of Stakeholder Management: A Systematic Review in the Construction Industry”, and discussed the impact of the digital transformation strategy on the relationship between customers and construction stakeholders (Line 182-187, page 8).

4.In the revision, we have re-edited the introduction, we focused on the urgent need for digital and service-oriented integration and stakeholder collaboration in the digital service ecosystem for the sustainable development of the construction industry and the mitigation of the negative impact of the COVID-19 epidemic and clarified the reality that this study is close to the needs of the industry (Line 69-90, page 4).

---

## [Editor Report · Decision Letter 1]

28 Apr 2023

Evolutionary game study on multi-agent value co-creation of service-oriented digital transformation in the construction industry

PONE-D-23-02979R1

Dear Dr. Su,

We’re pleased to inform you that your manuscript has been judged scientifically suitable for publication and will be formally accepted for publication once it meets all outstanding technical requirements.

Kind regards,

Simon Grima, PhD

Academic Editor

PLOS ONE

Additional Editor Comments (optional):

Reviewers' comments:

<quillbot-extension-portal></quillbot-extension-portal>

---

## [Editor Report · Acceptance letter]

5 May 2023

PONE-D-23-02979R1 

*Evolutionary game study on multi-agent value co-creation of service-oriented digital transformation in the construction industry*

Dear Dr. Su:

I'm pleased to inform you that your manuscript has been deemed suitable for publication in PLOS ONE. Congratulations! Your manuscript is now with our production department. 

Kind regards, 

on behalf of

Professor Simon Grima 

Academic Editor

PLOS ONE